# Improvement in the Electrochemical Properties of Lithium Metal by Heat Treatment: Changes in the Chemical Composition of Native and Solid Electrolyte Interphase Films

Paul Maldonado Nogales [ID], Hee-Youb Song, Mun-Hui Jo and Soon-Ki Jeong *[ID]

Department of Energy Systems Engineering, Soonchunhyang University, Soonchunhyang-ro 22-gil, Sinchang-myeon, Asan-si 31538, Korea; paul_mald@hotmail.com (P.M.N.); youbi0815@gmail.com (H.-Y.S.); munhui58@gmail.com (M.-H.J.)
* Correspondence: hamin611@sch.ac.kr; Tel.: +82-41-530-1313

**Abstract:** This study aims to improve the electrochemical properties of lithium metal for application as a negative electrode in high-energy-density batteries. Lithium metal was heat-treated at varying temperatures to modify the native and solid electrolyte interphase (SEI) films, which decreased the interfacial resistance between the lithium electrode and electrolyte, thereby improving the cycling performance. Moreover, the influence of the native and SEI films on lithium metals depended on the heat-treatment temperature. Accordingly, X-ray photoelectron spectroscopy (XPS) was performed to investigate the chemical composition of the native and SEI films on the heat-treated lithium metals before and after immersion in an organic electrolyte solution. The XPS results revealed the high dependence of the chemical composition of the outer layer of the native and SEI films on the heat-treatment temperature, implying that the native and SEI films can be effectively modified by heat treatment.

**Keywords:** lithium metal; heat treatment; native film; solid electrolyte interphase; X-ray photoelectron spectroscopy

## 1. Introduction

Lithium-ion batteries are widely used as power sources for portable devices, electric vehicles, and energy storage systems [1]. However, lithium-ion batteries employing graphite as a negative electrode cannot achieve a high energy density owing to the limited specific capacity of graphite (372 mAh·g$^{-1}$ and 841 mAh·cm$^{-3}$). In concept, the utilization of the lithium metal as a negative electrode can increase the energy density owing to its high theoretical specific capacity (3860 mAh·g$^{-1}$ and 2060 mAh·cm$^{-3}$) and low standard electrode potential ($-3.04$ V vs. standard hydrogen electrode) [2,3]. Nevertheless, lithium metal is yet to be used in commercial lithium secondary batteries because of its high reactivity under ambient atmospheric conditions and dendritic lithium deposition, which results in poor electrochemical performance, low cyclability, and low battery safety [4–8].

Although lithium metal exhibits high reactivity with various organic electrolyte solutions, lithium deposition can occur with the formation of a solid electrolyte interphase (SEI) that suppresses further decomposition of the electrolyte solution [9–11]. The SEI film is commonly formed on lithium metal in organic electrolyte solutions, acting as an electronic insulator and an ionic conductor, which enables reversible lithium deposition and dissolution at the lithium metal electrode. Generally, the electrochemical performance of the lithium metal electrode depends significantly on the nature of the SEI film [12–18]. Therefore, the formation of SEI film on the lithium metal electrode should be specifically controlled to improve the electrochemical performance, which is related to the reversibility and safety of lithium secondary batteries.

Moreover, a native film composed of various lithium compounds such as $Li_2O$, $Li_2CO_3$, and LiOH is commonly formed on lithium metals even in an inert atmosphere [19]. The native film on lithium metal affects the nature of the SEI film and the electrochemical properties of the electrode in organic electrolyte solutions [20–23]. Previous reports suggested that the native film should be modified to improve the electrochemical performance of lithium metal electrodes [19–24]. Therefore, various methods should be developed to modify native and SEI films on lithium metal electrodes and improve the electrochemical properties of the electrodes. Accordingly, in this study, a native film on lithium metal was modified by heat treatment, and the correlation between the components of the SEI films and the native films with heat treatments at temperatures of 26.8, 38.7, and 55.1 °C was investigated. Furthermore, a cycling performance test and impedance spectroscopy were performed to evaluate the electrochemical properties of the three types of lithium metals. In addition, X-ray photoelectron spectroscopy (XPS) was conducted to determine the chemical species of the native and SEI films formed on the lithium metal electrode. In the present study, we developed a new method of heat treatment to improve the electrochemical properties of lithium metal as a negative electrode.

## 2. Materials and Methods

### 2.1. Preparation of Lithium Metal at Various Heat-Treatment Temperatures and for Surface Analysis

Lithium metal purchased from VITZROCELL Co., Dangjin, Korea, was used to prepare three types of lithium metal electrodes by heat treatment at temperatures of 26.8, 38.7, and 55.1 °C. The heat-treatment temperature was controlled by varying the extruder speed in a dry room, wherein the dew point was maintained between −50 and −70 °C, as depicted in Figure 1. The elemental composition of the lithium ingot used for extrusion was $Li \geq 99.9$, $Na \leq 0.02$, $Ca \leq 0.02$, $Mg < 0.01$, $K \leq 0.01$, $Al \leq 0.005$, $Fe \leq 0.001$, $Cl \leq 0.006$, and $N \leq 0.02\%$. The heat-treatment temperatures were measured using a thermographic camera during the extrusion process. To form the SEI film, the lithium metal electrodes were immersed in 1 mol $dm^{-3}$ (M) of $LiPF_6$ dissolved in a 1:1 (*v/v*) mixture of ethylene carbonate (EC) and ethyl methyl carbonate (EMC) (Enchem, battery grade, Jecheon, Korea) for 7 days. Thereafter, the SEI-coated electrodes were washed in dimethyl carbonate in a glove box and dried for 10 h under vacuum at room temperature to remove the solvent. Subsequently, argon gas was injected into the vacuum drying chamber to obtain lithium metal electrodes for surface analysis. Surface analysis was then conducted for the electrodes using XPS (Thermo Fisher Scientific, K-Alpha, Waltham, MA, USA) under high vacuum ($5 \times 10^{-7}$ Pa). An Al Kα line was used as the X-ray source. An argon ion laser was sputtered for 10 min on the lithium surface for depth analysis. The prepared lithium metal was analyzed under XPS at various etching periods.

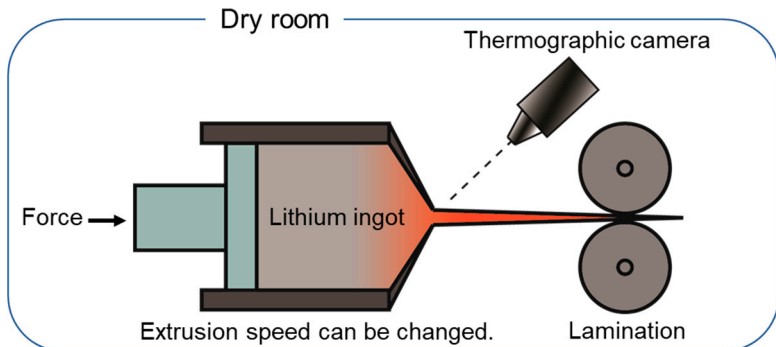

**Figure 1.** Schematic of preparation process of heat-treated lithium metal.

### 2.2. Electrochemical Measurements

The electrochemical properties of the lithium metal electrodes were investigated using a 2032-type coin cell. The cells for the cycling performance test were assembled with $Li[Ni_{0.33}Co_{0.33}Mn_{0.33}]O_2$ (NCM) + $LiMn_2O_4$ (LMO) (Wellcos, battery grade, Gunpo, Korea)

as the working electrode and lithium metal as the counter electrode. Lithium metal was heat-treated at various temperatures prior to the cell assembly. To manufacture the positive electrode, a slurry was prepared by mixing NCM + LMO (7:3, *w/w*) as an active material, poly(vinylidene chloride) (Sigma Aldrich, average MW 534000, St. Louis, MO, USA) as a binder, and carbon black (Super P) (Alfa Aesar, 99+%, Harverhill, MA, USA) as a conductive material in an *N*-methyl-2-pyrrolidone (Sigma Aldrich, 99+%, St. Louis, MO, USA) solvent for 3 h. The slurry was coated on an aluminum current collector using a blade-type coating machine (Rotech, AI-VR200, Gwangju, Korea) and dried at 120 °C in vacuum. The electrolyte solution used was 1 M $LiPF_6$/EC + EMC (1:1, *v/v*). The cells were galvanostatically charged and discharged at 0.4 C rate (95.6 mA·$g^{-1}$) between 3.0 and 4.2 V using a battery test system (Wonatech, WBCS3000, Seoul, Korea). A Li/Li symmetric cell was used for the electrochemical impedance spectroscopy (EIS), which was carried out at an open-circuit voltage in the frequency range of 100 kHz to 10 MHz with an alternating amplitude of 10 mV.

## 3. Results and Discussion

First, three types of lithium metal were prepared at three different heat-treatment temperatures: 26.8 °C (type A), 38.7 °C (type B), and 55.1 °C (type C). As depicted in Figure 2a, the interfacial resistance of the Li/Li symmetric cells with an equivalent circuit was measured after storage in 1 M $LiPF_6$/EC + EMC (1:1, *v/v*) for 7 days. In particular, the depressed arcs in the high- and mid-frequency regions could be attributed to the impedance of the lithium-ion migration through the SEI film and the charge-transfer reactions, respectively. Moreover, the equivalent circuit was typical, where $R_S$ denotes the electrolyte resistance, $R_{SEI}$ and $R_{CT}$ represent the resistances and $Q_{SEI}$ and $Q_{CT}$ denote the capacitances of the SEI film and charge-transfer reactions, respectively [25–27]. As the $R_{SEI}$ decreased with an increasing heat-treatment temperature, the nature of the native and SEI films can be presumed to vary with the three types of lithium metal electrodes. The cycling performance of the coin cells (Li[$Ni_{0.33}Co_{0.33}Mn_{0.33}$]$O_2$ + $LiMn_2O_4$/Li metal) was investigated to elucidate the correlation between the electrochemical properties and heat-treatment temperature (Figure 2b). The cycling performances of types A, B, and C over 60 cycles are comparatively presented in Figure 2b. In case of type B, the discharge capacity decreased most rapidly after 25 cycles, and a capacity retention of 9% was achieved after 60 cycles. However, the capacity retentions were 53% and 84% after 60 cycles for types A and C, respectively. These results revealed that the nature of the native and SEI films can be modified by the heat treatment of the lithium metal electrode, and the electrochemical performance could be significantly enhanced through appropriate heat treatment.

The XPS profiles of the Li 1s for the three types of lithium metal electrodes before immersion in the electrolyte solution are illustrated in Figure 3a–c. In particular, the peaks were obtained for various etching periods corresponding to the outer (0 min) and inner (10–50 min) sections of the native film. More specifically, the peaks at 52.3, 53.7, and 55.0 eV were assigned to Li, $Li_2O$, and $Li_2CO_3$, respectively [19–23]. The peaks corresponding to $Li_2O$ and $Li_2CO_3$ varied significantly for the types A, B, and C in the outer section. Primarily, $Li_2O$ was formed for types A and B, but $Li_2CO_3$ was formed for type C in case the heat-treatment temperature increased in the outer section. This indicated that the chemical composition of the outer section of the lithium metal electrode was affected by the heat-treatment temperature. However, similar lithium compounds were present in the inner section of the film, and peaks corresponding to $Li_2O$ and lithium metal were observed in all cases. Moreover, $Li_2O$ was mainly formed in the inner section for types A, B, and C. Thus, $Li_2O$ was primarily formed on the lithium metal and was influenced by the heat-treatment temperature.

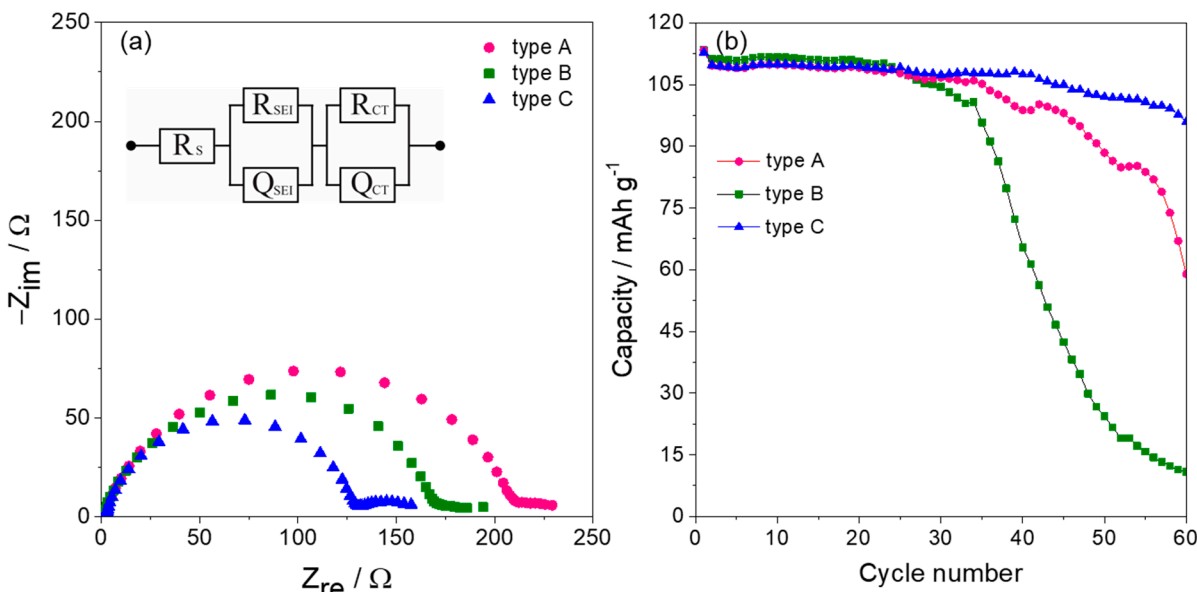

**Figure 2.** (**a**) Nyquist plots of the Li/Li symmetric cell with an equivalent circuit and (**b**) cycle performance of the 2032-type coin cells (NCM + LMO/Li metal) in 1 M LiPF$_6$/EC + EMC (1:1, *v/v*) after storing for 7 days.

The lithium compounds in the native film for types A, B, and C were evidently confirmed by the O 1s XPS profiles (Figure 3d–f). The peaks at 528.8, 531.6, and 532.0 eV correspond to Li$_2$O, LiOH, and Li$_2$CO$_3$, respectively [19–23]. Additionally, a high-intensity peak at 528.8 eV and a low-intensity peak at 531.6 eV were observed, indicating that Li$_2$O was primarily formed with a small amount of LiOH in the outer section for type A (Figure 3d). Moreover, a new peak at 532.0 eV was confirmed for both B and C (Figure 3e,f). The intensity of the peak corresponding to Li$_2$CO$_3$ increased with the heat-treatment temperature. For type B, Li$_2$O primarily existed in the outer section with a small amount of LiOH and Li$_2$CO$_3$. Eventually, Li$_2$CO$_3$ was formed as the main component, in addition to small amounts of Li$_2$O and LiOH for type C. Therefore, the composition of the outer section of the native film on lithium metal was significantly influenced by the heat-treatment temperature. Thus, heat treatment is an effective method for modifying native films on lithium metal.

The three lithium metal electrodes were immersed in 1 M LiPF$_6$/EC + EMC (1:1, *v/v*) for 7 d to form an SEI film. The XPS profiles of Li 1s for the SEI films of types A, B, and C electrodes are depicted in Figure 3a–c. After immersion, a strong peak at 56.0 eV attributed to LiF was observed in the outer section of the film of the electrodes caused by the reaction with the electrolyte solution [21,24]. Moreover, the formation of LiF was confirmed from the XPS profiles of F 1s (Figure S1), where LiF was formed as the major component in the outer section of the film of the electrodes after immersion. In addition, the peak at 56.0 eV was confirmed after etching for 40 or 50 min for types B and C (Figure 3b,c). As discussed above, the peak corresponding to Li$_2$CO$_3$ was observed in the native films of types B and C, which increased with the heat-treatment temperature. Therefore, Li$_2$CO$_3$ accelerated the formation of LiF in the outer and inner sections of the SEI film in the electrolyte solution. Based on the EIS and XPS results, the existence of LiF may result in a lower interfacial resistance of the lithium metal electrodes.

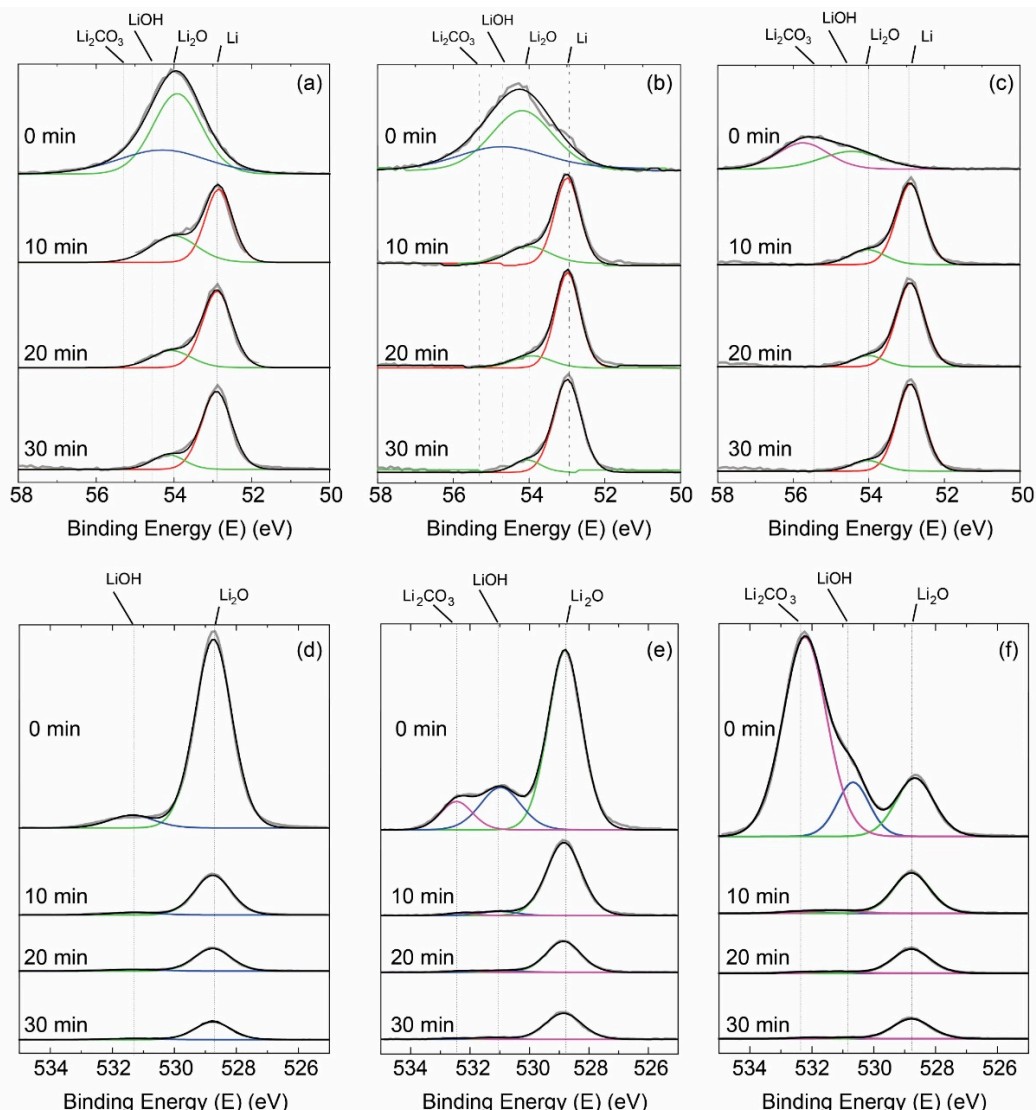

**Figure 3.** XPS profiles of (**a–c**) Li 1s and (**d–f**) O 1s for the heat-treated lithium metals at (**a,d**) 26.8, (**b,e**) 38.7, and (**c,f**) 55.1 °C before immersion.

The XPS profiles of O 1s for the SEI films of types A, B, and C electrodes are illustrated in Figure 4d–f. The peak corresponding to $Li_2CO_3$ was confirmed in the outer section of types A and C electrodes without any peaks related to $Li_2O$ and LiOH (Figure 4d,f). Consequently, LiF was mainly formed, and $Li_2CO_3$ was formed in small amounts in the outer section for all electrode types. However, the peaks corresponding to $Li_2O$, LiOH, and $Li_2CO_3$ were observed in the outer section of type B electrode (Figure 4e), suggesting the formation of LiF as the primary component along with the presence of small amounts of $Li_2O$, LiOH, and $Li_2CO_3$. As discussed above, type B displayed the lowest cycling performance, indicating that $Li_2O$ and LiOH in the outer section negatively impacted the electrochemical performance of the lithium metal electrode. The influence of the lithium compounds in the SEI film on the electrochemical properties of the electrode has not been clearly explained in this study, but it is evident that the lithium compounds in the native and SEI films are significantly influenced by the heat-treatment temperature. Although the SEI films of types A and C are structurally similar to each other, their chemical composition varied slightly. For type A, LiF was observed only in the outer section of the SEI film, whereas for type C, a small amount of LiF was observed in both the outer and inner sections of the SEI film, which may be related to the electrochemical performance of the lithium metal electrode. As reported, LiF—a component of SEI—produced a uniform

current distribution and resulted in a high degree of reversibility in lithium dissolution and deposition [22,23,28]. Therefore, the LiF-rich SEI film of type C may have resulted in the superior cycling performance, although direct correlations between the distribution of LiF and the suppression of dendritic lithium growth remain unclear. In general, organic compounds are formed on the lithium metal in electrolyte solution. Therefore, the effect of organic compounds on native and SEI films should be further investigated in future studies. The study demonstrates that the organic compounds beneficial for LiF formation hold research potential.

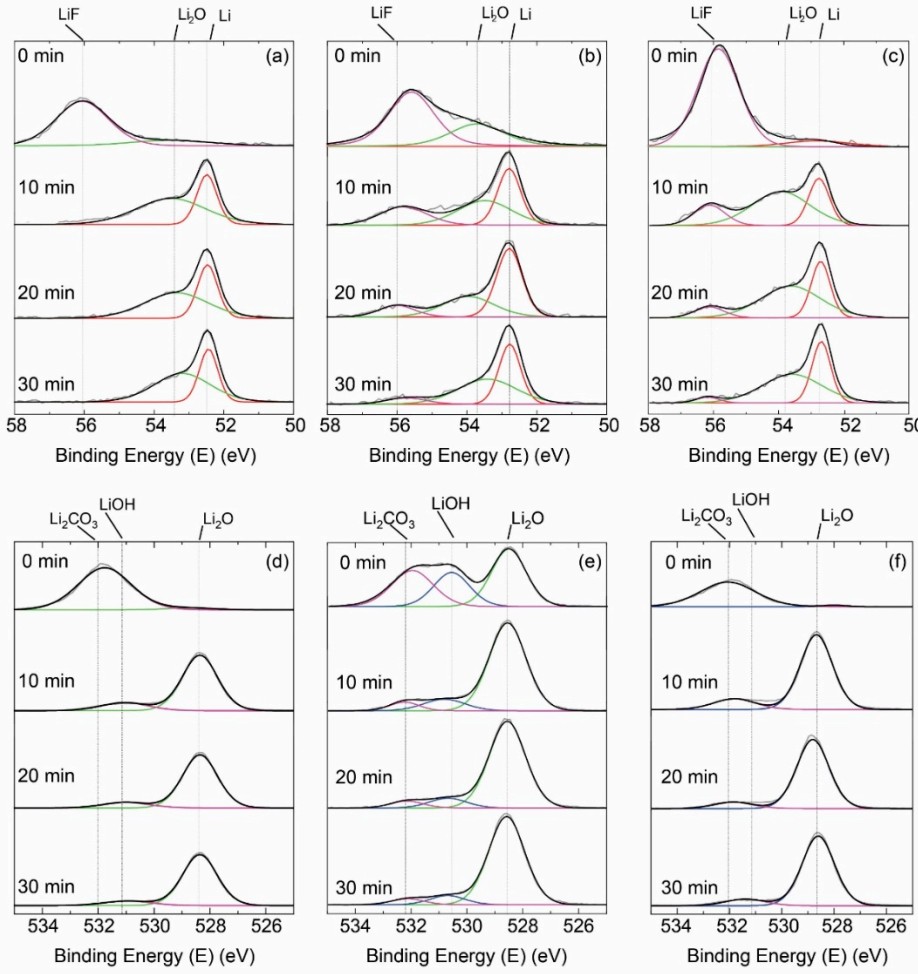

**Figure 4.** XPS profiles of (**a–c**) Li 1s and (**d–f**) O 1s for heat-treated lithium metals at (**a,d**) 26.8, (**b,e**) 38.7, and (**c,f**) 55.1 °C after immersion in 1 M LiPF$_6$/EC + EMC (1:1, $v/v$).

## 4. Conclusions

The native film of lithium metal was modified with the heat treatment, resulting in an effective SEI with a positive influence on the cycling performance of lithium metal electrode. The electrochemical properties and composition of native and SEI films were investigated for lithium metal electrodes by conducting heat treatments at 26.8, 38.7, and 55.1 °C. The results confirmed that the interfacial resistance of the heat-treated lithium metal electrodes decreased with the increasing heat-treatment temperature. In addition, the cycling performance improved after heat-treating the lithium metal electrode at 55.1 °C, implying that the performance of native and SEI films on lithium metal could be significantly enhanced through appropriate heat treatment. Moreover, the schematic models of the native and SEI films were obtained using XPS (Figure 5), which revealed that Li$_2$CO$_3$ was primarily formed in the outer section of native type C films, whereas Li$_2$O was mainly formed in the outer section for types A and B films. After immersion in the electrolyte solution, LiF

was the major compound formed in the outer section of the SEI film on all electrode types. In addition, a small amount of LiF was formed with $Li_2O$ as the fundamental component in the inner sections of the SEI film of types B and C. In case of type B, small amounts of $Li_2O$, LiOH, and $Li_2CO_3$ were present in the outer section of the SEI film. Thus, the composition of the outer section of the lithium metal electrode significantly varied with the heat-treatment temperature, but the inner section did not exhibit any such variation. Nonetheless, the detailed correlations between the composition and structure of the SEI film and the electrochemical properties of lithium metal electrodes remain unclear. Moreover, the three temperatures considered in this study could not adequately demonstrate the influence of the heat treatment, i.e., 55.1 °C cannot be considered as the optimum heat-treatment temperature. Nevertheless, we suggest an innovative strategy to form an effective SEI film by heat-treating lithium metal. The native film of lithium metal was modified by heat treatment, resulting in an effective SEI film with a positive influence on the cycling performance of the lithium metal electrode. The results of this study suggested that the composition of the inner component in contact with the lithium metal as well as that of the outer component in contact with the electrolyte solution is essential to determine the effectiveness of an SEI film. In future studies, the direct correlations between the heat-treatment temperatures and the electrochemical performance of lithium metal electrodes should be clarified by using more types of heat-treated lithium metal to identify the suitable heat-treatment temperature for the current purpose. Furthermore, the thickness of the SEI films on heat-treated lithium metals should be determined using various electron microscopes to explore various thicknesses of the film along with the positive/negative effects of the organic/inorganic compounds comprising them.

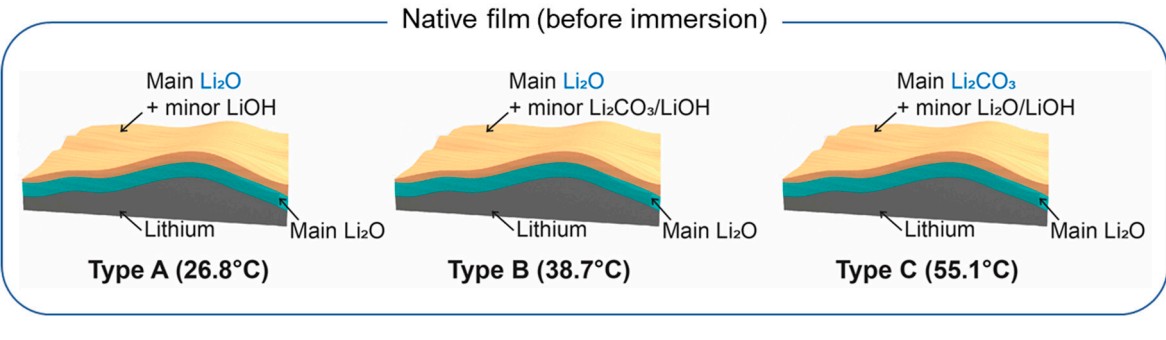

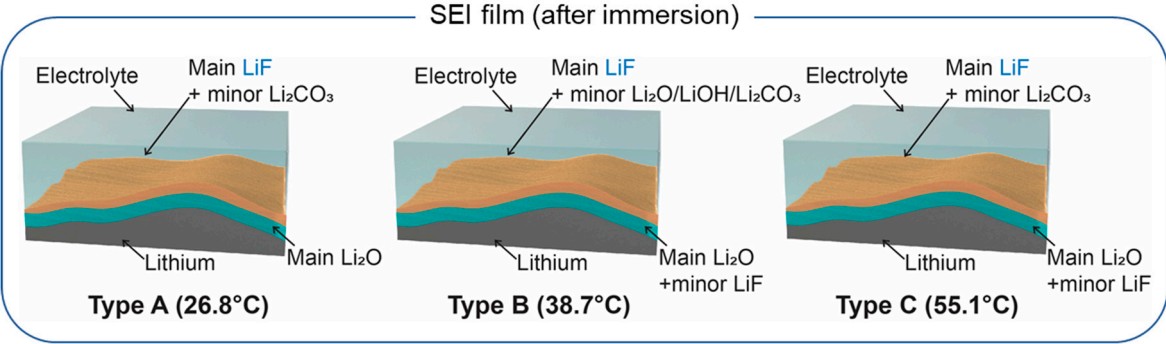

**Figure 5.** Schematic models of native and SEI films on heat-treated lithium metal at 26.8, 38.7, and 55.1 °C before and after immersion 1 M $LiPF_6$/EC + EMC (1:1, *v/v*).

**Supplementary Materials:** The following supporting information can be downloaded at: https://www.mdpi.com/article/10.3390/en15041419/s1, Figure S1: XPS F 1s results of heat-treated lithium metal after immersion.

**Author Contributions:** Conceptualization, H.-Y.S. and S.-K.J.; methodology, M.-H.J. and P.M.N.; formal analysis, H.-Y.S. and S.-K.J.; investigation, M.-H.J. and P.M.N.; resources, P.M.N.; data curation, P.M.N.; writing—original draft preparation, P.M.N. and S.-K.J.; writing—review and editing, S.-K.J.; supervision, S.-K.J.; project administration, S.-K.J.; funding acquisition, S.-K.J. All authors have read and agreed to the published version of the manuscript.

**Funding:** This research was funded by the Korea Institute of Energy Technology Evaluation and Planning (KETEP) and the Ministry of Trade, Industry & Energy (MOTIE) of the Republic of Korea (No. 20204030200060). This research was supported by Korea Electric Power Corporation (Grant number: R20XO02-25).

**Institutional Review Board Statement:** Not applicable.

**Informed Consent Statement:** Not applicable.

**Data Availability Statement:** Not applicable.

**Conflicts of Interest:** The authors declare no conflict of interest.

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
