# Peer review of "Improvement in the Electrochemical Properties of Lithium Metal by Heat Treatment: Changes in the Chemical Composition of Native and Solid Electrolyte Interphase Films"

_energies, doi:10.3390/en15041419_

Round 1
Reviewer 1 Report
The paper provided a simple but useful treatment method on Li metal that could potentially enhance the performance of Li-ion batteries using the treated Li metal as the counter electrode. Minor changes are needed, it is recommended for publication after considering the following comments:
Line 53 “… and was investigated” looks like has a grammar issue.
Section 2.1 Please make a schematic (flow chart) to show the process of extruding and heat treatment that would help the understanding and improve the paper a bit.
Section 2.1 and 2.2 are all chemicals coming from Enchem? Please make it clear.
Line 67 “dried for” I assume it is nature dried or is it under elevated temperature?
Line 68 “inert atmosphere” please be specific, like which inert gas is used...
Line 78 “The slurry was …” to make it flow smoothly, it could change to “To make the working electrode, a slurry was …”
Line 80-81, what is the coating method you used to coat the slurry?
Line 83, it is constant CHARGE mode rather than constant current, or the unit for 0.4C is 0.4A?
Line 97, the expression may course some misunderstanding, I suggest “Figure 1b shows the comparison among types A, B, and C of the cycling performance over 60 cycles”. The original expression may let the readers think that they differ after 60 cycles.
Line 102, “is significantly affected” I notice that you use “affected” throughout the paper, at least in some places like here, I suggest that you change to “could be significantly enhanced through proper heat treatment” to sell out your good findings.
Section 3 Figure 2. What kind of etching is used? Do you have an understanding of the etching rate on the surface? According to Figure 2, the surface does not change much after 10 minutes, so your outer section is etched out after less than 10 minutes, then the thickness of the modified layer could be obtained.
Figure 2, there is Li2CO3 formed on the surface, what is the carbon source? I assume it comes from the CO2 in the dry room. I suggest that you add the parameters of the dry room in section 2.1 (the gas component in the environment, humidity level,…)
Line 166 I suggest adding some sentences like “The study indicates that the organic compounds that are beneficial for LiF formation are worth being investigated” to further show your contribution in this paper. Please modify and insert the sentence according to the context.
Reviewer 2 Report
The manuscript focuses on investigation the effect of thermal treatment on electrochemical properties and composition of SEI films. The article is well-written and understandable. The manuscript will be interesting for the readership of the Energies.
However, there are some concerns that need to be addressed:
1) Please indicate the purity and suppliers for all the chemicals that were used in the study. For the Li metal please provide elemental analysis by ICP or provide specification from the manufacturer.
2) The coloured lines on the Figure 2 and Figure 3 is hardly visible – please increase the quality of the image (no less than 300 dpi) and increase thickness of the lines.
3) From your data, it can be concluded that the oxidation of the lithium electrode, leading to the formation of Li2O and LiOH, negatively affects the electrochemical properties, while an increased temperature favours Li + CO2 However after SEI film immersion into 1 M LiPF6/EC+EMC the composition of films of types A and C is practically identical (as it follows from Figure 4), so could you explain why they have so much difference in electrochemical properties?
Reviewer 3 Report
The paper has studied the electrochemical properties of lithium metal for application as a negative electrode in high-energy-density batteries, in particular the effect of thermal treatment on the solid electrolyte interphase (SEI) films.
Nevertheless, the results could represent the begin of an interesting results, the reported data are not enough to be published in Energies. Therephore, I suggest to reject the paper for the following reasons:
- It is not clear the novelty of the work and it is not reported relevant literature showing previous paper on the effect of lithium heat treatment on SEI properties;
- The experimental techniques and the number of samples used to characterize the electrode are poor to support the conclusion. Also, more temperature values should be analyzed to discuss a trend, only three values can be significant only if a high number of samples are used at the same conditions to verify reproducibility and analyze errors;
- EIS analysis should be discussed better by reporting relevant literature, model circuits used and physical meanings
- Figure 4 should be supported by SEM/TEM analysis
Round 2
Reviewer 3 Report
I suggest to move the sentence "“In the present study, we developed a new method of heat treatment to improve the
electrochemical properties of lithium metal as a negative electrode." to the end of the introduction section.
